# Night-time hot spring bathing is associated with improved blood pressure control: A mobile application and paper questionnaire study

Satoshi Yamasaki[1,2]*, Yusuke Kashiwado[1], Toyoki Maeda[1], Takahiko Horiuchi[1]

1 Department of Internal Medicine, Kyushu University Beppu Hospital, Beppu, Japan, 2 Department of Hematology and Clinical Research Institute, National Hospital Organization Kyushu Medical Center, Fukuoka, Japan

* yamas009@gmail.com

**Data Availability Statement:** All relevant data are within the manuscript and its Supporting information files.

## Abstract

Hot spring bathing practice helps to manage hypertension. However, the details of the relationship between hot spring bathing and hypertension remain unknown. Older people are thought to be less adept than younger people at using digital devices such as mobile applications. Whether mobile application questionnaires, which have been increasing in recent years, can be used by older people is unclear. To address the knowledge gap regarding the management of older patients with hypertension, we prospectively evaluated mobile application and paper questionnaires regarding night-time hot spring bathing in respondents who had a choice of which to use. Changes in blood pressure because of hot spring bathing were evaluated. To investigate the effects of night-time hot spring bathing on blood pressure in adults, 1116 volunteers at 14 institutions in Beppu completed the study, including 562 in the mobile application questionnaire group and 556 in the paper questionnaire group. A total of 474 of 477 (99.3%) respondents aged ≥65 years used paper questionnaires. There was a significantly lower drop in both systolic and diastolic blood pressure after using hot springs in respondents aged ≥65 years than in respondents aged <65 years ($p<0.001$). An age ≥65 years, hypertension with medication, arrhythmia, depression, and using a chloride hot spring were independently and significantly associated with a lower drop in both systolic and diastolic blood pressure after night-time hot spring bathing ($p<0.001$). Night-time hot spring bathing was significantly associated with reduced blood pressure in older adults ($p<0.001$). Extending this research by examining how psychosocial factors in respondents aged ≥65 years influence preferences for mobile and paper questionnaires may be beneficial, and further investigation is warranted.

## Introduction

The prevalence of hypertension in patients aged ≥20 years is high worldwide [1]. One study showed that the control of blood pressure to a systolic pressure of <130 mmHg and a diastolic

**Funding:** The author(s) received no specific funding for this work.

**Competing interests:** NO authors have competing interests.

pressure of <80 mmHg after antihypertensive drug therapy was achieved by 54%, 50%, 46%, and 33% of patients aged 20–54 years, 55–64 years, 65–74 years, and ≥75 years, respectively [2]. The treatment of hypertension should involve lifestyle modifications, such as dietary salt restriction [3], weight loss [4], better diet [5], exercise [6], limited alcohol intake [1], reduced smoking [7], reduced consumption of coffee and tea [8], more sleep [9], and hot spring bathing in older patients [10]. However, the details of the relationship between hot spring bathing and hypertension remain unknown.

In Japan, there is a long history of habitual daily bathing, which involves hot bathing rather than showering. In areas where there are hot springs, including Beppu, which contains the highest concentration of hot spring sources worldwide and has 9 of the 10 hot spring types, people appear to habitually prefer bathing in hot springs. Hot spring bathing, which has balneotherapeutic effects according to the hot spring type as reported by the Japanese Ministry of the Environment [11], is more effective for raising core body temperature than hot tap-water bathing.

Medical questionnaires, including those on hypertension [12], are widely administered in numerous medical fields and are used by many specialists. Using paper questionnaires is important to assess the effect of hot spring bathing on health-related issues, but previous questionnaire studies on this topic have been limited to small geographical areas, such as a single city [13]. The benefits of mobile application questionnaires include ease of use, immediate electronic storage of results and automatic scoring, avoidance of secondary data entry errors, and easier follow-up of records over time [14]. Additionally, there has been positive feedback on the use of mobile application questionnaires (which included informed consent procedures) to assess quality of life [15], medical education [16], interventions, diagnostics, and questionnaire completion [17].

To address the knowledge gap regarding the management of older patients with hypertension, we prospectively evaluated the use of mobile application and paper questionnaires regarding hot spring bathing and evaluated reductions in blood pressure owing to hot spring bathing. The primary objective of this study was to identify the type of questionnaires selected by adults aged ≥65 years. We also measured variables that could affect blood pressure, such as demographics, medical history, and concomitant medications. We performed univariate and multivariate regression analyses to examine the correlations between these variables. We identified which variables were dependent variables on the basis of our prespecified hypothesis of an inverse correlation between night-time hot spring bathing and blood pressure, as reported in a previous cross-sectional study [10].

## Materials and methods

### Study population

In this study, we developed a mobile application questionnaire from Neo Marketing Inc., Tokyo, Japan, and a paper (S1 Table) questionnaire. When completing electronic questionnaires on a mobile phone, an interface from Neo Marketing Inc. is sent to the mobile application questionnaires, while another interface from the mobile phone is sent to data storage in Neo Marketing Inc. This system enables adults to complete mobile application questionnaires directly on their mobile phone while enabling questionnaire researchers to quickly obtain all necessary questionnaires through the mobile application and analysis. The study population included 1116 adults aged ≥18 years who were using night-time hot springs at 14 institutions in Beppu (6 members of the Beppu Tourist Hotel Association, 7 hot springs organized by Beppu City, and our institution). Each participating institution has similar methods and

equipment to automatically measure blood pressure immediately before and after hot spring bathing.

We included 434 men and 684 women aged 22–94 years who met the inclusion criteria, provided consent, and were enrolled in this study between 26 June 2023 and 30 September 2023. The inclusion criteria were men and women aged at least 18 years who could read and who used night-time hot springs. In this study, the primary end point was that 80% of the respondents were able to complete the required questions. If 85% of respondents were able to complete the questions, the α error was 0.05, the power was 0.80, and n = 365 participants were required. Therefore, the required number of respondents was 400 for each form and application.

Initially, we used a questionnaire using a mobile application, but the respondents could also use a paper questionnaire if they preferred. This selection between mobile and paper question-naires, which was based on gender, age, and disease history, was used for further analysis of the data. All the data were collected and double-checked for accuracy by two separate research assistants. We examined the following variables: gender; age; hypertension with or without medication; and lifetime disease history, including arrhythmia, stroke, gout, diabetes mellitus, hyperlipidemia, renal disease, and chronic hepatitis, which are associated with a lower preva-lence of hypertension in our former study [10]. Informed consent for study participation was obtained by providing the respondents with information on mobile or paper questionnaires. The study was performed in accordance with the institutional guidelines and the principles of the Declaration of Helsinki. The protocol was approved by the institutional review board of Kyushu University Hospital, Japan (No. 22346–00).

## Statistical methods

The prespecified primary analyses of adult preference for a mobile phone or a paper form of the questionnaire used the chi-square test to determine whether the proportion of adults who considered a questionnaire by mobile phone to be the same, better, or much better than a paper version was significantly >50%. Similarly, adult preferences were analyzed in stratified groups on the basis of gender, age ($\geq$65 vs. <65 years), a history of disease, and type of hot spring as defined by the Japan Spa Association (https://www.spa.or.jp/en/spring-quality/; e.g., a "chloride spring" may include sodium chloride, calcium chloride, and magnesium chloride springs, depending on the main ingredients of the cation). We analyzed the frequencies and descriptive statistics of the variables. We excluded respondents if they had missing data for at least one of the covariates. Intergroup differences in categorical variables are expressed as the number and percentage. The chi-square test was used to examine the relationships between categorical variables.

Univariate and multivariate regression analyses were conducted to determine the associa-tions between the variables and a drop in systolic or diastolic blood pressure after night-time hot spring bathing. Covariates that were significant at $p<0.05$ in the univariate regression anal-ysis were included in the multivariate regression analysis. We prospectively calculated the coef-ficient and 95% confidence intervals of the effect of a change in systolic and diastolic blood pressure before and after using night-time hot springs using the multivariate regression model. A $p$ value <0.05 was considered statistically significant. All analyses were conducted using EZR (Saitama Medical Center, Saitama, Japan; http://www.jichi.ac.jp/saitama-sct/SaitamaHP.files/statmedEN.html) [18], which is a graphical user interface for R version 2.13.0 (www.r-project.org), and a modified version of R Commander version 1.6–3 designed to add statistical functions.

**Table 1. Characteristics of the survey respondents.**

| Characteristic | | Mobile<br>n = 562 | Paper<br>n = 556 | p |
|---|---|---|---|---|
| Gender, n (%) | Male | 158 (28.1) | 276 (49.6) | <0.001 |
| | Female | 404 (71.9) | 280 (50.4) | |
| Median age (range), years | | 48 (22–86) | 79 (60–94) | <0.001 |
| ≥65 years, n (%) | | 3 (0.5) | 474 (85.3) | <0.001 |
| Disease history, n (%) | | | | |
| Hypertension with medication | | 288 (51.2) | 452 (81.3) | <0.001 |
| Hypertension without medication | | 271 (48.2) | 84 (15.1) | <0.001 |
| Cancer | | 2 (0.4) | 102 (18.3) | <0.001 |
| Acute myocardial infarction and angina | | 3 (0.5) | 105 (18.9) | <0.001 |
| Arrhythmia | | 3 (0.5) | 94 (16.9) | <0.001 |
| Stroke | | 0 | 102 (18.3) | - |
| Gout | | 5 (0.9) | 51 (9.2) | <0.001 |
| Diabetes mellitus | | 17 (3.0) | 254 (45.7) | <0.001 |
| Hyperlipidemia | | 112 (19.9) | 220 (39.6) | <0.001 |
| Renal disease | | 3 (0.5) | 153 (27.5) | <0.001 |
| Depression | | 391 (69.6) | 187 (33.6) | <0.001 |
| Collagen disease | | 1 (0.2) | 0 | - |
| Hot spring bathing, n (%) | | | | |
| Hot spring type | | | | |
| Simple | | 98 (17.4) | 72 (12.9) | <0.001 |
| Chloride | | 464 (82.6) | 484 (87.1) | 0.044 |
| Time | | | | |
| 19:00 to 20:00 | | 307 (54.6) | 457 (82.2) | <0.001 |
| 20:00 to 21:00 | | 215 (38.3) | 53 (9.5) | |
| 21:00 to 22:00 | | 34 (6.0) | 46 (8.3) | |
| 22:00 or 23:00 | | 6 (1.1) | 0 | |
| Median (range) BP before bathing, mmHg | | | | |
| Systolic BP | | 156 (150–159) | 154 (140–159) | <0.001 |
| Diastolic BP | | 93 (87–94) | 94 (75–98) | <0.001 |
| Median (range) BP after bathing, mmHg | | | | |
| Systolic BP | | 145 (112–147) | 126 (112–149) | <0.001 |
| Diastolic BP | | 85 (65–89) | 75 (64–88) | <0.001 |
| Median (range) drop in BP after bathing, mmHg | | | | |
| Systolic BP | | 11 (9–44) | 28 (3–47) | <0.001 |
| Diastolic BP | | 8 (1–26) | 17 (3–31) | <0.001 |

The p values were obtained using the chi-square test.

BP, blood pressure.

## Results

The baseline characteristics of the respondents who used night-time hot springs are shown in Table 1. The median (range) age of the respondents who used night-time hot springs in the mobile application and paper questionnaire groups was 48 (22–86) and 79 (60–94) years, respectively. The respondents in the mobile application questionnaire group were younger than those in the paper questionnaire group ($p < 0.001$).

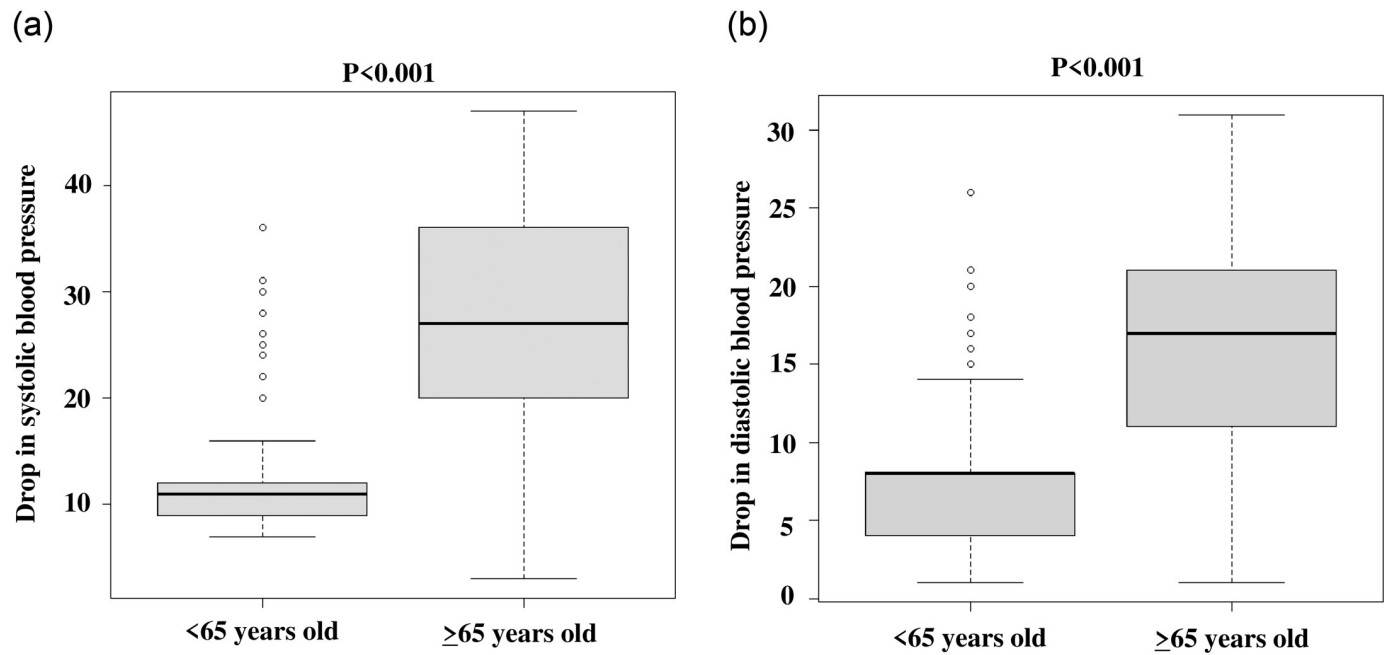

**Fig 1.** a. Drop in systolic blood pressure after night-time hot spring bathing. b. Drop in diastolic blood pressure after night-time hot spring bathing.

To evaluate the association between age (≥65 vs. <65 years old) and a drop in systolic or diastolic blood pressure, we compared the drop in systolic (median: 27 vs. 11 mmHg, range: 3–47 vs. 7–36 mmHg, respectively) and diastolic (median: 17 vs. 8 mmHg, range: 1–31 vs. 1–26 mmHg, respectively) blood pressure after using a night-time hot spring (Fig 1a and 1b, respectively). The drop in both systolic and diastolic blood pressure was significantly lower in respondents aged ≥65 years than in respondents aged <65 years ($p<0.001$). No complications, side effects, or adverse events were reported from the 14 participating institutions during this study.

In the multivariate regression analysis, male gender ($p<0.001$), age ≥65 years ($p<0.001$), a disease history including hypertension with medication ($p<0.001$), hypertension without medication ($p = 0.007$), arrhythmia ($p<0.001$), gout ($p<0.001$), and depression ($p<0.001$), and using a chloride hot spring ($p<0.001$) were significantly associated with a drop in systolic blood pressure after night-time hot spring bathing (Table 2). An age ≥65 years ($p<0.001$), a disease history including hypertension with medication ($p<0.001$), arrhythmia ($p<0.001$), and depression ($p<0.001$), and using a chloride hot spring ($p<0.001$) were significantly associated with a drop in diastolic blood pressure after night-time hot spring bathing (Table 3).

In the multivariate regression analysis, a disease history, including acute myocardial infarction and angina ($p<0.001$), diabetes mellitus ($p<0.001$), and hyperlipidemia ($p = 0.004$), was inversely associated with a drop in systolic blood pressure after night-time hot spring bathing (Table 2). A disease history, including cancer ($p<0.001$), acute myocardial infarction and angina ($p<0.001$), diabetes mellitus ($p<0.001$), hyperlipidemia ($p<0.001$), and renal disease ($p<0.001$), was inversely associated with a drop in diastolic blood pressure after night-time hot spring bathing (Table 3).

**Table 2. Univariate and multivariate regression analyses of variables affecting the drop in systolic blood pressure after night-time hot spring bathing.**

| Parameter | Variable | Univariate | | | | Multivariate | | | |
|---|---|---|---|---|---|---|---|---|---|
| | | Coefficient | 95% CI | s.e. | p | Coefficient | 95% CI | s.e. | p |
| Gender | Male vs. Female | 2.113 | 0.975, 3.251 | 0.580 | <0.001 | 2.098 | 0.960, 3.237 | 0.580 | <0.001 |
| Age | ≥65 years vs. <65 years | 12.697 | 11.465, 13.929 | 0.627 | <0.001 | 13.103 | 11.960, 14.245 | 0.582 | <0.001 |
| Disease history | Hypertension with medication | 6.215 | 3.117, 9.313 | 1.578 | <0.001 | 6.060 | 2.966, 9.154 | 1.577 | <0.001 |
| | Hypertension without medication | 4.381 | 1.214, 7.547 | 1.613 | 0.006 | 4.319 | 1.151, 7.487 | 1.614 | 0.007 |
| | Cancer | −0.342 | −2.137, 1.452 | 0.914 | 0.708 | | | | |
| | Acute myocardial infarction and angina | −12.041 | −15.107, −8.976 | 1.562 | <0.001 | −10.722 | −13.412, −8.033 | 1.370 | <0.001 |
| | Arrhythmia | 11.861 | 9.379, 14.343 | 1.265 | <0.001 | 10.961 | 8.678, 13.244 | 1.163 | <0.001 |
| | Gout | 6.595 | 3.135, 10.055 | 1.763 | <0.001 | 5.577 | 2.384, 8.770 | 1.627 | <0.001 |
| | Diabetes mellitus | −5.008 | −6.394, −3.622 | 0.706 | <0.001 | −5.210 | −6.566, −3.854 | 0.691 | <0.001 |
| | Hyperlipidemia | −1.746 | −2.891, −0.600 | 0.583 | 0.002 | −1.537 | −2.597, −0.477 | 0.540 | 0.004 |
| | Renal disease | −1.457 | −3.065, 0.150 | 0.819 | 0.075 | | | | |
| | Depression | 3.697 | 2.562, 4.832 | 0.578 | <0.001 | 3.638 | 2.518, 4.758 | 0.570 | <0.001 |
| Hot spring type | Chloride vs. Simple | 3.760 | 2.546, 4.974 | 0.618 | <0.001 | 3.872 | 2.663, 5.080 | 0.615 | <0.001 |

s.e., standard error; CI, confidence interval.

Univariate or multivariate competing event statistics analyzed using a regression model were applied to a reduction in systolic blood pressure after night-time hot spring bathing. Covariates significant at $p<0.05$ in the univariate analysis were included in the multivariate analysis.

## Discussion

This study showed that 99.3% of respondents aged ≥65 years used paper questionnaires. There was a significantly lower drop in both systolic and diastolic blood pressure after using hot springs in respondents aged ≥65 years than in respondents aged <65 years. An age ≥65 years, a disease history including hypertension with medication, arrhythmia, and depression,

**Table 3. Univariate and multivariate regression analyses of variables affecting the drop in diastolic blood pressure after night-time hot spring bathing.**

| Parameter | Variable | Univariable | | | | Multivariable | | | |
|---|---|---|---|---|---|---|---|---|---|
| | | Coefficient | 95% CI | s.e. | p | Coefficient | 95% CI | s.e. | p |
| Gender | Male vs. Female | −0.185 | −0.803, 0.432 | 0.315 | 0.556 | | | | |
| Age | ≥65 years vs.<65 years | 5.366 | 4.697, 6.035 | 0.341 | <0.001 | 5.377 | 4.713, 6.042 | 0.338 | <0.001 |
| Disease history | Hypertension with medication | 2.585 | 0.902, 4.268 | 0.857 | 0.002 | 1.455 | 0.912, 1.997 | 0.276 | <0.001 |
| | Hypertension without medication | 1.147 | −0.572, 2.867 | 0.876 | 0.190 | | | | |
| | Cancer | −2.216 | −3.191, −1.240 | 0.496 | <0.001 | −1.971 | −2.908, −1.035 | 0.477 | <0.001 |
| | Acute myocardial infarction and angina | −7.351 | −9.016, −5.685 | 0.848 | <0.001 | −6.361 | −7.499, −5.223 | 0.579 | <0.001 |
| | Arrhythmia | 10.021 | 8.672, 11.369 | 0.687 | <0.001 | 9.474 | 8.322, 10.627 | 0.587 | <0.001 |
| | Gout | 1.716 | −0.1633, 3.595 | 0.957 | 0.073 | | | | |
| | Diabetes mellitus | −6.876 | −7.629, −6.124 | 0.383 | <0.001 | −7.089 | −7.809, −6.368 | 0.367 | <0.001 |
| | Hyperlipidemia | −2.861 | −3.483, −2.239 | 0.317 | <0.001 | −2.791 | −3.396, −2.186 | 0.308 | <0.001 |
| | Renal disease | −3.422 | −4.295, −2.548 | 0.445 | <0.001 | −3.206 | −4.046, −2.365 | 0.428 | <0.001 |
| | Depression | 1.478 | 0.862, 2.095 | 0.314 | <0.001 | 1.464 | 0.941, 1.987 | 0.266 | <0.001 |
| Hot spring type | Chloride vs. Simple | 2.016 | 1.357, 2.676 | 0.336 | <0.001 | 2.087 | 1.431, 2.743 | 0.334 | <0.001 |

s.e., standard error; CI, confidence interval.

Univariate or multivariate competing event statistics analyzed using a regression model were applied to diastolic blood pressure reduction after night-time hot spring bathing. Covariates significant at $p<0.05$ in the univariate analysis were included in the multivariate analysis.

and using a chloride hot spring were independently and significantly associated with a drop in both systolic and diastolic blood pressure after night-time hot spring bathing.

In this prospective study, adults aged ≥65 years preferred to use paper questionnaires rather than mobile application questionnaires. Studies on electronic questionnaires using tablet computers reported no difficulty in most patients, except those who were older, those with less than a high school education, and those with comorbid medical conditions [19,20]. Similar to previous studies [20], we found that mobile questionnaires were favored by younger patients with a higher education level. Previous research shows that patients who use a mobile device daily appear to be more likely to prefer mobile questionnaires than those who do not [20]. It would be useful to extend this research to adults aged ≥65 years by examining how psychosocial factors such as self-efficacy, as well as physical factors such as visual or memory impairment, affect preferences for mobile and paper questionnaires. We believe that improving the ease of use and feasibility of mobile questionnaires for adults aged ≥65 years (e.g., by including video explanations of how to complete the questionnaire and appropriate consent acquisition procedures) might help to increase respondents' satisfaction and preference for mobile questionnaires over paper questionnaires.

To date, only one study has investigated the association between habitual night-time hot spring bathing and a lower prevalence of hypertension [10]. However, several previous studies have demonstrated the benefits of traditional thermal therapy, including hot spring bathing, for hypertension [21–23]. The present study showed that night-time hot spring bathing resulted in a drop in both systolic and diastolic blood pressure. The reduction in both systolic and diastolic blood pressure after night-time bathing was significantly greater in respondents aged ≥65 years than in those aged <65 years. Tai *et al.* reported that Japanese hot water bathing, especially in the short time from the end of bathing to bedtime, was associated with lower night-time and sleep-time blood pressure and greater dipping in an older adult population [24]. Therefore, night-time hot spring bathing to reduce blood pressure may be more suitable for older adults than for younger adults.

The present study showed that hypertension with medication or arrhythmia was independently and significantly associated with a drop in both systolic and diastolic blood pressure after night-time hot spring bathing. This finding indicates the need for specific intervention strategies aimed at older adults living with hypertension or arrhythmia, one of which could be night-time hot spring bathing. Systolic blood pressure is considered the major predictor of coronary disease in older patients [25]. An elevation in systolic blood pressure is primarily due to diminished arterial compliance, and isolated systolic hypertension may result from an increase in cardiac output owing to anemia, aortic insufficiency, and arteriovenous fistula [26]. However, Protogerou *et al.* have suggested that lower diastolic blood pressure is associated with an increased risk of coronary heart disease in older adults [27]. The rapid drops in blood pressure after hot spring bathing may be detrimental for some patients. There is also evidence that low prestroke blood pressure is associated with mortality after stroke, particularly among patients with at least one comorbidity, such as smoking, heart disease, cancer, or dementia [28]. Understanding the effect of hot spring bathing on reducing the risk of coronary heart disease is important and requires further research; clarification of this association may be particularly relevant for assessing the reduced riskT of coronary heart disease in older patients after hot spring bathing. In this study, we did not follow the baseline blood pressure because of the difficulty of data collection. To evaluate whether the antihypertensive effects of hot spring bathing are persistent, we are currently conducting a prospective, longitudinal study in hospitalized patients with hypertension.

In the multivariate regression analysis, we found that depression was independently and significantly associated with a drop in both systolic and diastolic blood pressure after night-

time hot spring bathing. We found that night-time hot spring bathing, which can improve sleep disorders owing to depression, may be associated with a drop in both systolic and diastolic blood pressure in older patients. In a large-scale study of an older population, taking a night-time hot spring bath shortened sleep onset latency if bathing was scheduled 1–3 hours before bedtime [27]. Brief exposure to sauna bathing can result in benefits for <1 hour, such as reduced blood pressure and improved arterial stiffness. However, sauna bathing for ≥3 weeks with repeated frequency can upregulate certain enzymes and pathways, which results in greater stress tolerance, an enhanced cellular environment, and improved health [23]. Owing to lack of physical activity and healthy nutrition in older patients with depression, practical interventions to prevent or improve hypertension, such as habitual night-time hot spring bathing, warrant additional attention.

Our study showed that hot spring bathing in a chloride spring bath was independently and significantly associated with a drop in both systolic and diastolic blood pressure after night-time hot spring bathing. Therefore, people with hypertension may benefit from taking a chloride spring bath. The effect of sodium chloride concentrations on physical and oxidative stability has been shown to be useful in cooking [28], and a clinical trial showed that sodium chloride was highly efficient in increasing core body temperature [29]. Additionally, a small Japanese study suggested that hot spring bathing in water containing sodium chloride maintained a higher rate of heat stability and was more efficient in elevating core body temperature than bathing in hot spring water without sodium chloride [29]. However, these findings should be confirmed in a prospective study with a control group, which is outside of the scope of the present study.

There were some study limitations. First, some selection bias was expected in this prospective study, which was performed at 14 institutions. In this study, bias was present owing to the following differences in data selection and other factors: a lack of data on respondents who engaged in hot spring bathing for the treatment of various diseases; respondents' income, which might be correlated with some vascular diseases or the frequency of hot spring bathing; respondents' lifestyle and diet, such as food consumption and obesity, sodium intake, drinking and smoking habits, consumption of coffee and tea, physical activity, and sleep; and the time at which respondents engaged in hot spring bathing. To minimize bias, we limited the inclusion criteria to patients with data on gender, age, disease history, and use of night-time hot spring bathing. Second, no long-term data on blood pressure were available. Therefore, further studies are required to assess the details of the long-term outcomes of patients who undergo hot spring bathing. Third, additional studies are needed that examine the effect of hot spring bathing while controlling for concurrent lifestyle and medication changes. Forth, some patients with primary or secondary hypertension may have been overlooked or the changes in blood pressure underestimated because of antihypertensive drug therapy. Additionally, there were no data specific to hot spring bathing, such as the duration of immersion, frequency, temperature, time, and years of habitual hot spring bathing. Finally, there is a lack of generalizability of the findings of this study to other populations.

## Conclusions

Although the use of mobile questionnaires in numerous medical fields has increased, we found that 99.3% of respondents aged ≥65 years chose to use paper questionnaires. Mobile questionnaires have the potential to increase clinical efficiency, improve physician—patient communication, and improve the patient's encounter overall. Therefore, the ease and feasibility of mobile questionnaires may help to increase respondents' preference for these questionnaires. This study showed a significantly lower drop in both systolic and diastolic blood

pressure after using hot springs in respondents aged ≥65 years than in respondents aged <65 years. Night-time hot spring bathing was significantly associated with reduced blood pressure in older adults. Extending this research by examining how psychosocial factors may affect preferences for mobile and paper questionnaires could be beneficial, and further investigation is warranted.

## Supporting information

**S1 File. File shows survey raw in S1 Table data.** Age, Q2; Age65over, over 65 years old; Disease, Q3 disease history; Hot spring chro, Q4 hot spring type; Time in and Time19, Q5 start time; PreSBP in, Q6 systolic blood pressure before hot spring bathing; PreDBP in, Q6 diastolic blood pressure before hot spring bathing; PostSBP out, Q6 systolic blood pressure after hot spring bathing; PostDBP out, Q6 diastolic blood pressure after hot spring bathing; SBP delta, drop in systolic blood pressure; DBP delta, drop in diastolic blood pressure; App, application. (XLSX)

**S1 Table. English version of the Japanese questionnaire used in this study.** (DOCX)

## Acknowledgments

We thank the respondents and clinical staff for their participation in the study. We appreciate the help of Mr. Yasuhiro Nagano (Mayor of Beppu), the Tourist Division of Beppu City, and the Beppu Tourist Hotel Association, especially Hotel Sunbury Annex, Hanabeppu, and Amane Resort, for enabling data collection. We thank Dr. Ellen Knapp, PhD, and Diane Williams, PhD, from Edanz (https://jp.edanz.com/ac) for editing a draft of this manuscript.

## Author Contributions

**Conceptualization:** Satoshi Yamasaki, Yusuke Kashiwado.

**Data curation:** Satoshi Yamasaki.

**Formal analysis:** Satoshi Yamasaki.

**Funding acquisition:** Satoshi Yamasaki.

**Investigation:** Satoshi Yamasaki.

**Methodology:** Satoshi Yamasaki, Yusuke Kashiwado.

**Project administration:** Satoshi Yamasaki, Yusuke Kashiwado.

**Resources:** Satoshi Yamasaki.

**Software:** Satoshi Yamasaki.

**Supervision:** Satoshi Yamasaki, Toyoki Maeda, Takahiko Horiuchi.

**Validation:** Satoshi Yamasaki.

**Visualization:** Satoshi Yamasaki.

**Writing – original draft:** Satoshi Yamasaki.

**Writing – review & editing:** Satoshi Yamasaki, Yusuke Kashiwado, Toyoki Maeda, Takahiko Horiuchi.

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
