## [Decision Letter · Decision Letter 0]

9 Jan 2024

PONE-D-23-34165Night-time hot spring bathing is associated with improved blood pressure control: A mobile application and paper questionnaire studyPLOS ONE

Dear Dr. YAMASAKI,

Thank you for submitting your manuscript to PLOS ONE. After careful consideration, we feel that it has merit but does not fully meet PLOS ONE’s publication criteria as it currently stands. Therefore, we invite you to submit a revised version of the manuscript that addresses the points raised during the review process. Please refer to the reviewer questions and revision recommendations. 

We look forward to receiving your revised manuscript.

Kind regards,

Vikramaditya Samala Venkata

Academic Editor

PLOS ONE

Journal Requirements:

https://www.auajournals.org/doi/10.1016/j.juro.2018.03.117

In your revision ensure you cite all your sources (including your own works), and quote or rephrase any duplicated text outside the methods section. Further consideration is dependent on these concerns being addressed.

"NO authors have competing interests"

4. In the online submission form, you indicated that [Insert text from online submission form here]. 

Reviewers' comments:

Reviewer's Responses to Questions

**Comments to the Author**

1. Is the manuscript technically sound, and do the data support the conclusions?

Reviewer #1: Yes

Reviewer #2: Partly

Reviewer #3: Partly

2. Has the statistical analysis been performed appropriately and rigorously? 

Reviewer #1: Yes

Reviewer #2: Yes

Reviewer #3: I Don't Know

3. Have the authors made all data underlying the findings in their manuscript fully available?

Reviewer #1: Yes

Reviewer #2: Yes

Reviewer #3: Yes

4. Is the manuscript presented in an intelligible fashion and written in standard English?

Reviewer #1: Yes

Reviewer #2: Yes

Reviewer #3: Yes

5. Review Comments to the Author

Reviewer #1: This is a very good and interesting study that shows effects of hot spring bathing in different salt and how it can lower Blood pressure . The statistical analysis was drawn in multiple individuals male and female with and without additional comorbidities- overall I enjoyed reading the study it does engage the audience and there are some minor revisions as suggested in the PDF attached

Reviewer #2: - Its not clear ,if the hypertension was primary or secondary hypertension

- Appx 50% of the population in the study were using medications for control of Blood pressure

- Timing of the repeat blood pressure check after the hot spring bath was not clear . Was repeat blood pressure checked immediately following the bath or the next day

- Effect of decreased blood pressure was it transient ? persistent in patients who were not using Anti Hypertensive medications ?

- Were there any complications /side effects or adverse events during the study

Reviewer #3: Interesting study question and innovative study design with both mobile and paper questionnaires. I have a few thoughts

1. There is some discrepancy between the table and the discussion regarding the numbers. Especially the median blood pressure drop post-spring bathing.

2. May be worth mentioning how the BP was recorded pre and post-spring bath. This is important, as lack of uniformity would affect the study results directly.

3. The claimed magnitude of BP drop, especially in the> 65-year age group, is significant and has not been shown with any other intervention in the past. But, as mentioned this data may not be generalized. Lack of data regarding concurrent lifestyle and medication changes limits the validity of this finding.

4. Spelling check especially in the tables.

5. I commend the authors for the additional information on mobile vs paper questionnaire. This definitely gives prespective for future research involving different age groups.

6. Please explain the inverse relationship of some risk factors with blood pressure drop in detail.

6. PLOS authors have the option to publish the peer review history of their article (what does this mean?). If published, this will include your full peer review and any attached files.

Reviewer #1: **Yes: **Dr Gurpreet Kaur Saini

Reviewer #2: No

Reviewer #3: **Yes: **ramprakash devadoss

---

## [Author Response · Author response to Decision Letter 0]

22 Jan 2024

Dr. Vikramaditya Samala Venkata

Academic Editor

PLOS ONE

15 January 2024

Re: Manuscript no. PONE-D-23-34165

Dear Dr Venkata,

We would like to thank you and the reviewers for the constructive comments on our manuscript entitled, “Night-time hot spring bathing is associated with improved blood pressure control: A mobile application and paper questionnaire study.” We have revised the manuscript in accordance with these comments; revisions to the text are shown in yellow highlight. Our point-by-point responses to the comments are provided below.

We hope that our manuscript will now be considered suitable for publication in PLOS ONE. We look forward to hearing from you at your earliest convenience.

Yours sincerely,

Satoshi Yamasaki, MD, PhD

Department of Internal Medicine

Kyushu University Beppu Hospital

4546 Tsurumihara, Tsurumi, Beppu, Oita 874-0838, Japan

Tel: +81-977-27-1600; Fax: +81-977-27-1641

E-mail: yamas009@gmail.com

Review Comments to the Author

Reviewer #1: This is a very good and interesting study that shows effects of hot spring bathing in different salt and how it can lower Blood pressure . The statistical analysis was drawn in multiple individuals male and female with and without additional comorbidities- overall I enjoyed reading the study it does engage the audience and there are some minor revisions as suggested in the PDF attached

RESPONSE: Thank you for your thoughtful and constructive comments on our manuscript; we appreciate your help. In accordance with your suggestions, we have revised the relevant text in the manuscript (page 2, line 18; page 3, line 45; page 5, lines 76–77; page 17, lines 252–253 and line 257).

Reviewer #2: - Its not clear ,if the hypertension was primary or secondary hypertension

RESPONSE: Thank you for raising this issue. We acknowledge that more details about the type of hypertension are needed. We have added a mention of these factors to the Limitations (page 19, line 287).

- Appx 50% of the population in the study were using medications for control of Blood pressure

RESPONSE: Thank you for your constructive comments on our manuscript. As you mention, 51.2% in the mobile group and 81.3% in the paper group of the study population were using hypertension medications. This may be owing to our promotion of the preventive effect of high blood pressure by engaging in night-time hot spring bathing (Reference No. 10). We acknowledge that participants’ use of antihypertensive medication may have made it difficult to clarify the effect of hot spring bathing on blood pressure. We have mentioned this factor as a study limitation (page 19, lines 285–286).

- Timing of the repeat blood pressure check after the hot spring bath was not clear . Was repeat blood pressure checked immediately following the bath or the next day

RESPONSE: To address your question, we have added some text explaining this point to the Material and Methods (page 6, lines 97–98).

- Effect of decreased blood pressure was it transient ? persistent in patients who were not using Anti Hypertensive medications ?

RESPONSE: Thank you for raising this issue. We did not have data on the long-term effects of hot spring bathing. We acknowledge the need for long-term information about changes in blood pressure, and have mentioned this factor as a study limitation (page 19, lines 283–285).

- Were there any complications /side effects or adverse events during the study

RESPONSE: Thank you for raising this issue. We acknowledge the need for more detail about the complications, side effects, and adverse events during the study. We have added a mention of these factors to the Results (page 10, lines 165–166).

Reviewer #3: Interesting study question and innovative study design with both mobile and paper questionnaires. I have a few thoughts

1. There is some discrepancy between the table and the discussion regarding the numbers. Especially the median blood pressure drop post-spring bathing.

RESPONSE: We apologize for the unclear text. We have revised the Abstract, Results, and Discussion to clarify these data (page 2, line 31; page 3, line 35; page 10, line 163; page 13, lines 192; page 14, line 196; page 15, lines 217, 218 and 226; page 17, lines 248, 250 and 262; page 20, line 301). We hope that this has clarified the data interpretation. However, we welcome your comments on any specific aspects of the data presentation you feel are still unclear.

2. May be worth mentioning how the BP was recorded pre and post-spring bath. This is important, as lack of uniformity would affect the study results directly.

RESPONSE: In accordance with your suggestion, we have added an explanation of this to the Material and Methods (page 6, lines 97–98).

3. The claimed magnitude of BP drop, especially in the> 65-year age group, is significant and has not been shown with any other intervention in the past. But, as mentioned this data may not be generalized. Lack of data regarding concurrent lifestyle and medication changes limits the validity of this finding.

RESPONSE: Thank you for your helpful comments. We acknowledge that this study has several limitations, including the effect of various unmeasured factors that correlate with blood pressure, such as lifestyle and diet (food consumption and obesity), sodium intake, drinking and smoking habits, consumption of coffee and tea, physical activity, and sleep. We have mentioned these as study limitations in the Discussion (page 18, lines 278–280; page 19, lines 285–286). 

4. Spelling check especially in the tables.

RESPONSE: We apologize for any errors. We have revised the tables to ensure that the spelling is correct (pages 9–10, line 155; pages 13–14, lines 180–181).

5. I commend the authors for the additional information on mobile vs paper questionnaire. This definitely gives perspective for future research involving different age groups.

RESPONSE: Thank you for your comment. In the Discussion, we have briefly expanded on the potential of mobile questionnaires for adults aged ≥65 years (pages 14–15, lines 209–211).

6. Please explain the inverse relationship of some risk factors with blood pressure drop in detail.

RESPONSE: To address your query, we have added some text explaining this point to the Discussion (page 16, lines 233–239).

---

## [Editor Report · Decision Letter 1]

5 Feb 2024

Night-time hot spring bathing is associated with improved blood pressure control: A mobile application and paper questionnaire study

PONE-D-23-34165R1

Dear Dr. YAMASAKI,

We’re pleased to inform you that your manuscript has been judged scientifically suitable for publication and will be formally accepted for publication once it meets all outstanding technical requirements. 

Kind regards,

Vikramaditya Samala Venkata

Academic Editor

PLOS ONE
---

## [Editor Report · Acceptance letter]

27 Mar 2024

PONE-D-23-34165R1 

PLOS ONE

Dear Dr. YAMASAKI, 

I'm pleased to inform you that your manuscript has been deemed suitable for publication in PLOS ONE. Congratulations! Your manuscript is now being handed over to our production team.

Kind regards, 

on behalf of

Dr. Vikramaditya Samala Venkata 

Academic Editor

PLOS ONE